# Robot-Assisted Training for Upper Limb in Stroke (ROBOTAS): An Observational, Multicenter Study to Identify Determinants of Efficacy

**DOI:** 10.3390/jcm10225245

**Published:** 2021-11-11

**Authors:** Rocco Salvatore Calabrò, Giovanni Morone, Antonino Naro, Marialuisa Gandolfi, Vitalma Liotti, Carlo D’aurizio, Sofia Straudi, Antonella Focacci, Sanaz Pournajaf, Irene Aprile, Serena Filoni, Claudia Zanetti, Maria Rosaria Leo, Lucia Tedesco, Vincenzo Spina, Carmelo Chisari, Giovanni Taveggia, Stefano Mazzoleni, Nicola Smania, Stefano Paolucci, Marco Franceschini, Donatella Bonaiuti

**Affiliations:** 1IRCCS Centro Neurolesi Bonino-Pulejo P.O Piemonte, Viale Europa, 98124 Messina, Italy; roccos.calabro@irccsme.it; 2IRCCS Fondazione Santa Lucia, Via Ardeatina, 306/354, 00179 Roma, Italy; g.morone@hsantalucia.it (G.M.); s.paolucci@hsantalucia.it (S.P.); 3Stroke Unit, Policlinico Universitario Messina, 98124 Messina, Italy; g.naro11@alice.it; 4Department of Neurosciences, Biomedicine and Movement Sciences, University of Verona, Via S. Francesco, 22, 37129 Verona, Italy; marialuisa.gandolfi@univr.it (M.G.); nicola.smania@univr.it (N.S.); 5UOC Medicina Fisica e Riabilitativa ASL di Pescara, 65124 Pescara, Italy; vitalma.liotti@ausl.pe.it (V.L.); carlo.daurizio@ausl.pe.it (C.D.); 6Neuroscience and Rehabilitation Department, Ferrara University Hospital, 44121 Ferrara, Italy; s.straudi@ospfe.it; 7Riabilitazione Intensiva Neurologica, S.C. Medicina Fisica e Riabilitazione Ospedaliera ASL4 Liguria, 16034 Sestri Levante, GE, Italy; antonella.focacci@asl4.liguria.it; 8Department of Neurological and Rehabilitation Sciences, IRCCS San Raffaele Roma, 00163 Rome, Italy; sanaz.pournajaf@sanraffaele.it; 9IRCCS Fondazione Don Carlo Gnocchi ONLUS, Via di Scandicci 269, 50143 Florence, Italy; iaprile@dongnocchi.it; 10Fondazione Centri di Riabilitazione Padre Pio ONLUS, Presidio “Gli Angeli di Padre Pio”, Viale Padre Pio, 24, 71013 San Giovanni Rotondo, Italy; serena.diba@gmail.com; 11Presidio Ospedaliero Accreditato Casa di Cura Villa Bellombra, Via Bellombra, 24, 40136 Bologna, Italy; claudia.zanetti@villabellombra.it (C.Z.); mrosaria.leo@villabellombra.it (M.R.L.); 12Physical and Rehabilitation Medicine, San Gerardo Hospital, 20900 Monza, Italy; l.tedesco@asst-monza.it; 13SD Neuroriabilitazione, Azienda Ospedaliero, Universitaria Pisana, 56121 Pisa, Italy; vincenzospina90@yahoo.it (V.S.); c.chisari@ao-pisa.toscana.it (C.C.); 14Habilita Istituto Clinico di Riabilitazione, Via P. A. Faccanoni, 6, 24067 Sarnico, Italy; giovannitaveggia@habilitasarnico.it; 15Department of Electrical and Information Engineering, Politecnico di Bari, Via Edoardo Orabona, 4, 70126 Bari, Italy; stefano.mazzoleni@poliba.it; 16Italian Society of Physical Medicine and Rehabilitation, Italy; dbonaiuti2@yahoo.it

**Keywords:** robot-assisted therapy, stroke, exoskeleton, rehabilitation, upper limp therapy

## Abstract

Background: The loss of arm function is a common and disabling outcome after stroke. Robot-assisted upper limb (UL) training may improve outcomes. The aim of this study was to explore the effect of robot-assisted training using end-effector and exoskeleton robots on UL function following a stroke in real-life clinical practice. Methods: A total of 105 patients affected by a first-ever supratentorial stroke were enrolled in 18 neurorehabilitation centers and treated with electromechanically assisted arm training as an add-on to conventional therapy. Both interventions provided either an exoskeleton or an end-effector device (as per clinical practice) and consisted of 20 sessions (3/5 times per week; 6–8 weeks). Patients were assessed by validated UL scales at baseline (T0), post-treatment (T1), and at three-month follow-up (T2). The primary outcome was the Fugl-Meyer Assessment for the upper extremity (FMA-UE). Results: FMA-UE improved at T1 by 6 points on average in the end-effector group and 11 points on average in the exoskeleton group (*p* < 0.0001). Exoskeletons were more effective in the subacute phase, whereas the end-effectors were more effective in the chronic phase (*p* < 0.0001). Conclusions: robot-assisted training might help improve UL function in stroke patients as an add-on treatment in both subacute and chronic stages. Pragmatic and highmethodological studies are needed to confirm the showed effectiveness of the exoskeleton and end-effector devices.

## 1. Introduction

Currently, various robotic systems have been implemented for the rehabilitation of the upper limb (UL) in stroke outcomes [1,2]. The use of robots for UL training has increasingly spread over the past three decades, and they are becoming common in the neurorehabilitation setting. However, there is not a consensus regarding the clinical indications and training protocols. Robotic therapy provides task-oriented, repetitive, high-intensity, and highly reproducible sensorimotor training to boost stroke plasticity-dependent recovery [1,2]. Generally speaking, robotic UL devices provide augmented feedback and performance dueto a 2D monitor that increases subjects’ attention, enjoyment, and motivation [3].

Several guidelines have recommended robotic UL training for rehabilitation after stroke to promote training intensity [4]. A recent Cochrane review concluded that people receiving electromechanical and robot-assisted training have more chances to improve their daily living activities, arm function, and arm muscle strength, with high-quality evidence [5]. However, the variation of the training characteristics and the type of training and the absence of specific outcome measures limit the applicability of evidence. The characterization of the type of patient that could benefit from the treatment with different robotic systems, in terms of neurological severity, type, and site of cerebral stroke; time onset; and type of deficit (i.e., sensitives and sensory-motor and cognitive deficits) remains an aspect of extreme clinical relevance that, to date, has beenpoorly explored [2,6,7].

A recent multicenter RCT (namely RATULS) regarding end-effector devices recently concluded that robot-assisted arm training was not superior to usual care for patients with moderate or severe UL functional limitation. This results in the need to determine whether thereis a use for robot-assisted training in routine clinical practice [8].

Understanding the aspects concerning the proper use of robotics could either have positive repercussions on clinical practice by proposing customized pathways to promote the maximum recovery possible or improve the scientific research sector to study the recovery mechanisms promoted by specific technological devices.

From an engineering perspective, robotic/electromechanical devices can be classified into end-effectors and exoskeletons. These two mechanical options reflect two different training approaches with a different balance between the freedom of motor control (end point vs. single joint), potentially leading to different functional recovery mechanisms. Unlike the lower limb, the UL function (i.e., shand grasping) needs specific generated devices (given that UL movements on different freedom degrees are complex, and especially those of the hand, which are not easy to emulate and reproduce), and this complicates the robotics’ approach to UL rehabilitation [2,9].

This study aimed to explore in real-life clinical practice the effect of robot-assisted training using end-effector and exoskeleton robotic devices on upper limb function following a stroke, attempting to draw specific indication of the use of both devices.

## 2. Materials and Methods

### 2.1. Participants

The study was conceived as a prospective, multicenter, observational cohort study involving 18 neurorehabilitation centers in Italy with experience in UL robot-assisted training and comparable organization setting. All of the inpatients were assessed by validated UL scales at baseline (T0), post-treatment (T1), and at three months’ follow-up (T2). The T2 follow-up was performed after discharge. Three hundred and fifty patients affected by acute and chronic post-stroke were consecutively screened for study inclusion from January 2017 to December 2019.

The inclusion criteria were (a) first-ever supratentorial ischemic cerebrovascular event documented with neuroradiological examinations (CT, MRI); (b) a clinically evaluated functional UL impairment, with sensorimotor and monoparesis/hemiparesis deficits; (c) age > 18 years; and (d) ability to maintain the sitting position. The exclusion criteria were (a) bilateral UL impairment; (b) cognitive or behavioral impairment that affects the understanding or execution of robotic training; (c) impossibility or unavailability to provide informed consent; (d) treatment with botulinum toxin in the previous three months and throughout the study (including follow-up); and (e) other medical severe problems potentially interfering with the training.

The study was approved by the Ethics Committee of the leading center (S. Gerardo Hospital, ASST Monza, Italy). All participants gave written informed consent before study participation. The study was registered with the number ACTRN12620000029998.

The enrolled patients were stratified in each center according to disease onset (early subacute onset: <30 days, late subacute onset: 30–60 days, and chronic onset: >60 days) [10,11], the level of impairment (severe, moderate, and mild, according to the baseline value of the FMA-UE), and age (younger than 50 years; between 50 and 70 years; older than 70 years) to have clinicallymatched groups. No allocation was provided, as the study aimed to evaluate the real use of UL robots in clinical practice.

### 2.2. Rehabilitation Paradigm

The affected UL was trained at either distal or proximal level using one of the assigned devices per clinical use/availability. Indeed, participants would have received the intervention regardless of this study.

The intervention with the exoskeleton devices (i.e., Armeo Power and Armeo Spring) included 20 sessions of 40 min (including 10 min for setup), 3 or 5 times per week for 6–8 weeks (according to their current clinical practice). Exoskeletons allow functional movements assisted by robots at the shoulder, elbow, and wrist with video feedback in an exergaming/2D-VR context. The robot assistance was progressively reduced according to subject improvement/capacity.

The intervention with the end-effector devices (i.e., InMotion 2, Armotion, Motore, and ReoGo) included the same amount of training, although the mechanism by which UL was treated was different. Table 1 shows the main features of the robots used in our clinical rehab settings. Overall, each participant was provided with one-hour daily UL robotic training regardless of the robot type. In each session, the training could involve the shoulder, elbow, and wrist joints when possible. The tasks to be performed were adapted to the workspace and difficulty consistently with the subject’s residual ability. The therapeutic task focused on functional improvement, including task-oriented exercises, sensorimotor reorganization, and spasticity inhibition. Subjects performed passive, active, and active-assisted exercises on UL joints to improve joint function, prevent contractures, inhibit spasticity, and improve motor function. A trained physical therapist supervised the robot-aided training sessions at each center as an add-on to conventional UL treatment (e.g., stretching, passive movements, and active movements, including pushing and punching) consistent with the Italian national guidelines [10].

### 2.3. Outcome Measures

The primary outcome measure was the rate of achievement of the minimal clinically important difference (MCID) in FMA-UE at the T1–T0 change, which consisted in the change from baseline of the FMA-UE of at least 10 points in the subacute phase [10] and of at least 5 points in the chronic one [11] after the treatment, in both groups. Specifically, we measured the effectiveness of between-group and within-group intervention as the number of patients who achieved the MCID at the end of the end-effector treatment compared to the exoskeleton one. In addition, the effects of the factors time since injury and level of impairment on FMA-UE were assessed.

The secondary outcomes consisted of the change from T0 to T1 in Motricity Index (MI), Box & Block Test (B&B), Numerical Rating Scale (NRS), Frenchay Arm Test (FAT), Barthel Index (BI), and Modified Ashworth Scale (MAS).

Finally, we investigated the possible baseline predictors of motor recovery (including age, time since injury, and baseline impairment).

### 2.4. Statistical Analysis

The significance of the changes in each outcome measure was calculated by conducting an analysis of variance for repeated measures (RM ANOVA) with time (three levels: T0, T1, and T2) and group (two levels: end-effector and exoskeleton) as factors. In addition, we compared the changes in the primary outcome measure from T0 to T1 using the factors time since injury (early-subacute, late-subacute, and chronic) [11,12] and the level of impairment (severe, moderate, and mild, according to the baseline value of the FMA-UE) [13] as covariates in the main analysis. For all statistical tests, the significance level was set at α < 0.05. Depending on the significance of the main interactions and effects, pairwise comparisons with Bonferroni correction were tested. Specifically, t-test calculation was conducted in each robotic subgroup to estimate intragroup effects. The values of all clinical outcome measures recorded between T0 and T1 (baseline and post-treatment, respectively), and T0 and T2 (baseline and three-month follow-up, respectively) were analyzed with the Wilcoxon rank test in the SigmaStat environment in the case of measurements deriving from ordinal scales and with t-tests for continuous numerical data. The statistical significance was set at *p* < 0.05. Finally, we categorized the subjects as improved or not improved according to the achievement of the MCID for FMA-UE in each time since injury category, and then we conducted a multivariable logistic regression with the clinical/demographic features as candidate predictors of post-treatment recovery (including age, time since injury, and baseline impairment). The odds ratio (OR) was calculated for each intervention. All the analyses were conducted according to a modified intention-to-treat analysis, including all participants for which data were available.

## 3. Results

The baseline characteristics (T0) are summarized in Table 2. There were no significant clinical/demographic differences between the groups.

### 3.1. Primary Outcome Measure

Patients’ experimental flowchart is reported in Figure 1.

RM ANOVA returned a significant time × group interaction (F = 77, *p* < 0.0001, λ = 155; time effect *p* < 0.0001). FMA-UE improved after the treatment by 7 points (30%) on average in the end-effector group (time effect *p* < 0.0001) and by 11 points (60%) on average in the exoskeleton group (time effect *p* < 0.0001)(Table 3).

Notably, the observed improvements in FMA-UE significantly depended on the time since injury factor (time × group F = 2.9, *p* = 0.008, λ = 18) and the level of impairment (F = 9.6, *p* = 0.0001, λ = 29).

When comparing the changes in the primary outcome measure from T0 to T1 using the factor time since injury (early-subacute, late-subacute, and chronic) (Figure 2), the FMA-UE, the percentage of patients achieving the MCID in the exoskeleton group was 32% early-subacute (+18 points), 20% late-subacute (+21 points), and 40% chronic patients (+7 points), whereas the percentage of patients achieving the MCID in the end-effector group was 21% early-subacute (+11 points), 18% late-subacute (+11 points), and 57% chronic patients (+16 points). T1–T0 comparisons were significant (all *p* < 0.0001) only in all patients grouped together and in the improved patients (i.e., those who achieved the MCID). Therefore, the increase in the exoskeleton group was higher than the MCID (i.e., at least 10 points) and was substantially appreciable in the early- and late-subacute patients, whereas the FMA increase in the end-effector group was substantially higher than the MCID (i.e., at least 5 points) in the chronic patients (Figure 2). Exoskeletons were indeed more effective in the early- (OR = 4.63, *p* < 0.0001) and late-subacute phases (OR = 27, *p* < 0.0001) than in the chronic one, where the end-effectors were more effective (OR = 20, *p* < 0.0001). However, the FMA-UE improvement was not comparable between the groups when adjusted for the baseline FMA-UE (*p* < 0.0001) and the time since injury (*p* < 0.0001). Therefore, the primary outcome measure was sufficient to detect a difference between the two groups.

When comparing the changes in the primary outcome measure from T0 to T1 using the factor baseline level of impairment, the FMA-UE improvement was significant (higher than the MCID) in the patients provided with end-effectors with mild (both time intervals *p* = 0.004) and moderate impairment (both time intervals *p* = 0.002) compared to those with severe impairment (both time intervals *p* = 0.2) (Figure 3). Conversely, the FMA-UE improvement was significant (higher than the MCID) in the patients provided with exoskeletons with severe (both time intervals *p* = 0.002) impairment compared to those with mild and moderate functional impairment (both time intervals *p* = 0.2) (Figure 3). Exoskeletons were thus more effective in the more severe patients (OR = 2.66, *p* = 0.002), whereas end-effectors offered better results in the mild-to-moderate patients (OR = 1.9, *p* = 0.02).

Each robot yielded significant improvement in FMA-UE, but the most evident changes were appreciable following ArmeoPower in the exoskeleton group and InMotion^®^ 2.0 in the end-effector group (Table 4).

Lastly, age (*t* = 54, *p* < 0.001), time since injury (*t* = 35, *p* < 0.001), and baseline impairment (as per FMA) (*t* = 36, *p* < 0.001) were the variables predicting recovery (MCID achievement) according to the logistic regression analysis.

### 3.2. Secondary Outcome Measures

The time×groupinteraction was significant for B&B (*p* = 0.006), MAS-shoulder (*p* = 0.02), and MI (*p* < 0.001), whereas BI, MAS-elbow, MAS-wrist, NRS, and FAT were not significantly affected (Table 3). The effect of baseline severity (as per FMA-UE) was significant only for B&B (*p* = 0.003) and BI (*p* = 0.02), whereas time since injury did not affect any secondary outcome measure.

## 4. Discussion

The present multicenter observational trial aimed to explore the effect of robot-assisted training by means of end-effector and exoskeleton robotic devices on UL function following stroke in real-life clinical practice. Robotic UL therapy performed with either an end-effector or an exoskeleton was feasible in subjects with chronic and subacute stroke, as shown by a large body of literature [5,14]. Both robot-assisted therapies improved UL function, as suggested by the achievement of the MCID independently of the time onset. This finding is in accordance with the current literature and a recent meta-analysis analyzing 2654 subjects [7]. On the other hand, we found that the higher the degree of impairment, the more effective the robotic arm therapy was. In particular, exoskeletons were more effective in more severe patients, whereas end-effectors offered better results in mild-to-moderate patients, regardless of the time since injury. These innovative and interesting results represent the first step to fulfillingthe requirement of reaching clear evidence on selecting specific types of robotic devices for specific subgroups of patients according to their severity, sensory deficits, and the objective of the robotic-assisted training.

These results are in line with those previously demonstrated in lower limb robotic therapy in post-stroke individuals [15] and with a recent systematic review with a meta-analysis concluding the superiority of robotic therapyover conventional therapy. Notably, such a superiority is evident in the more severe subjects, who likely have limited potential for spontaneous recovery [16].

Consistently with this finding, the study by Duret et al. [17] highlighted that patients more severely affected might benefit more from robotic therapy than conventional UL therapy. On the contrary, it has been shown that robotic treatment significantly improved UL motor function, activities, and participation to the same extent as a similar amount of conventional therapy in subjects with subacute stroke [18]. This is why the field deserves further investigation, including evidence from clinical settings.

Recovery from a stroke event is a complex process entailing both spontaneous and motorpractice-mediated processes. Partial structural and functional impairments likely recover through a potentiation and extension of residual brain areas, whereas complete lesions of specific brain areas require a substitution by functionally related systems [19,20,21]. It is fundamental to be aware of such processes and related outcomes to better understand when to expect recovery, plan the most appropriate treatment, and determine the timing of rehabilitation. Although it is widely recognized that spontaneous behavioral recovery mostly occurs within the first three months after stroke, different patterns of recovery may then emerge depending on many complex factors [19]. Indeed, chronic stroke patients may still experience functional recovery through promoting cerebral plasticity, as evaluated by transcranial magnetic stimulation, EEG [22], and advanced neuroimaging techniques [23].

The natural response to disability is to learn new ways of accomplishing daily activities by developing compensatory behaviors, beyond the restoration of the acquired ones. Stroke survivors with UL deficits typically learn to rely on the non-paretic hand and arm, leading to the “learned nonuse” phenomenon and, thus, exacerbating impairments [24]. Moreover, weakness, sensory impairments, and pain can prevent the “normal” movement of the affected UL when it is forced to move; therefore, compensatory strategies are used to complete the task. Thus, another possible explanation for robotic-related recovery could be the learned nonuse phenomenon, which is more likely to occur in the more severe stroke subjects during conventional therapy [25]. This promising hypothesis should be investigated in a well-planned RCT.

Regarding the type of robot (exoskeleton or end-effector), although there was not a clear superiority of one device type over the other, different robot-assisted devices should be used in different stroke phases in the real-life setting. End-effectors hold the patient’s hand or forearm at one point and generate forces at the interface. Their system designs the trajectories that match the hand’s natural trajectory in space for the required task, and the joints do not match those of the UL. Consequently, these devices do not allow UL intersegmental control.

For this reason, end-effectors might be suitable for patients with residual motor skills sufficient to control their movement and could be employed after exoskeleton robots in rehabilitation settings [6].

Contrariwise, exoskeletons should be used in more severe patients and in the acute phase as suggested by the literature data [16]. This issue deserves the following considerations: (1) exoskeleton devices are more recent and less diffused but with high potentiality as shown by our results; (2) the high potentiality of exoskeleton regards the possibility to treat even more severe subjects in a 3D workspace due to the high degree of constriction that they might apply in comparison with end-effector devices; (3) our results showed that subacute subjects might have a greater improvement with exoskeletons, and this could be explained by the fact that impairment and motor control deficits are more important in the early stage of recovery, and a higher degree of constriction is desirable during the treatment. From another side, end-effector devices are generally less expensive, and there is more clinical experience with their use.

The main strength of our study is the explorationof the effects of different robot-assisted devices used in the real-life neurorehabilitation setting as an add-on treatment for UL rehabilitation. This approach allows the collection of relevant information to plan future studies with a higher methodology (i.e., RCT).

Our research has several limitations: the observational design and the absence of a dose-matched conventional group limit the generalization of our results regarding the superiority of robotic therapy. Nonetheless, the assessment of superiority was not the aim of the present study. A mitigation strategy of this condition was applied to evaluate the MCID. There was a lack of longerterm follow-up to detect the time the UL recovery can be considered complete. There was also a lack of rigid standardization of the interventions implemented. Finally, our study lacked a cost/effectiveness analysis to compare device and personnel cost differences.

## 5. Conclusions

Robot-assisted UL therapy might improve motor function in post-stroke patients regardless of the time since injury, as well as in chronic patients. Patients with a more severe deficit benefited more from robotic UL therapy than those with a less severe deficit, depending on the employed device. These data reflect the need for UL training with the use of more objective, flexible, and controlled therapeutic paradigms.

## Figures and Tables

**Figure 1 jcm-10-05245-f001:**
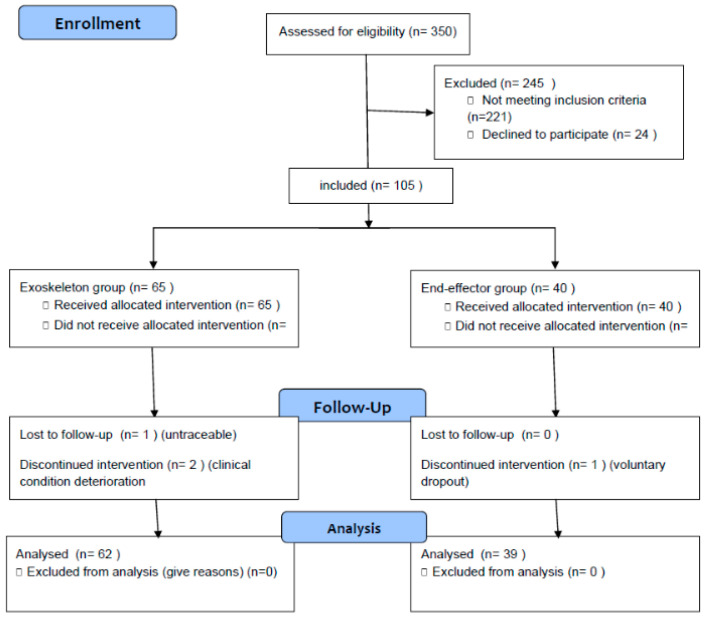
Patients’ flow diagram.

**Figure 2 jcm-10-05245-f002:**
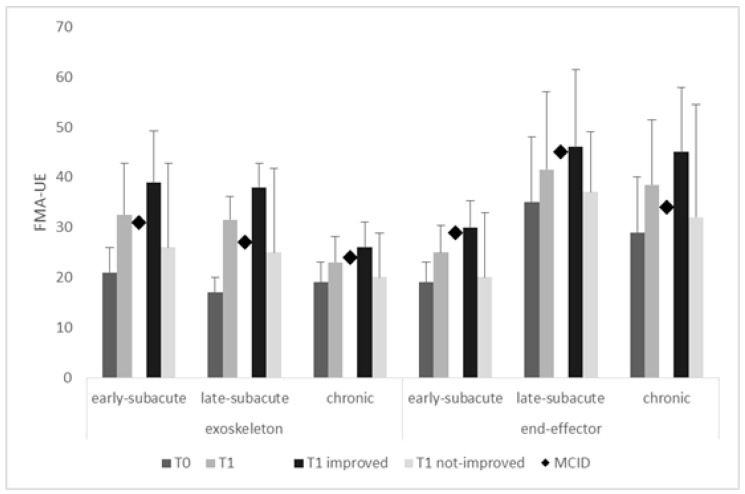
Fugl-Meyer Assessment for the upper extremity (FMA-UE) in the different subgroups at T0 and T1 (data from all the participants as well as from improved and notimproved patients). T1–T0 comparisons were significant (all *p* < 0.0001) only in all patients grouped together and in the improved patients (i.e., those who achieved the MCID). The vertical error bars refer to SD. Minimal clinically important difference, MCID.

**Figure 3 jcm-10-05245-f003:**
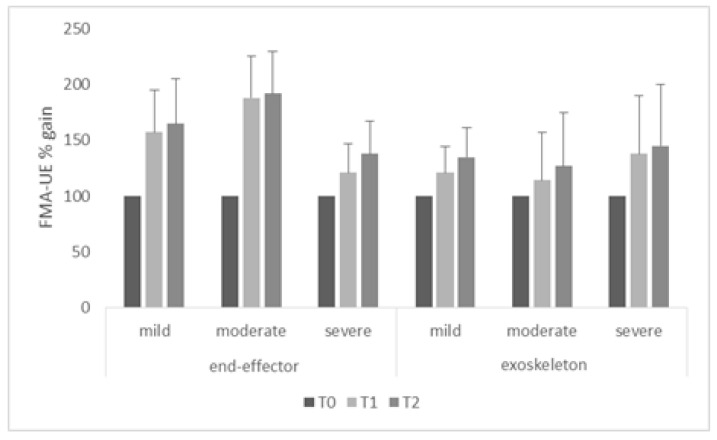
Post-treatment Fugl-Meyer Assessment for the upper extremity (FMA-UE) gain (i.e., at T1 and T2 with respect to the level of impairment of the baseline (T0) FMA-UE value, calculated as T1/T0 × 100 and T2/T0 × 100) in both groups and from all the patients. The vertical error bars refer to SD.

**Table 1 jcm-10-05245-t001:** Summary of the features of the devices employed (device classification and brand name) for upper limb rehabilitation.

Exoskeleton	Armeo Power (Hocoma AG, Switzerland)	The device has been specifically designed for arm and hand therapy in an early stage of rehabilitation. It enables even patients with severe movement impairments to perform exercises with high repetitions (high intensity), which is paramount for relearning motor function. It uses sensors and intelligent algorithms to recognize when the patient cannot carry out movement in a 3D space and assists the patient’s arm as much as needed to successfully reach the goal of the exercise in a 2D exergaming.
	Armeo Spring (Hocoma AG, Switzerland)	By providing arm weight support, the Armeo Spring enables patients to use any remaining motor functions and encourages them to achieve a higher number of reach and grasp movements based on specific therapy goals in a 3D space working environment. All activity during the training is based on the patient’s movements. An extensive library of game-like augmented performance feedback exercises have been designed to train core movement patterns commonly used in daily living activities.
End-effector	InMotion 2.0 (Bionik Laboratories, Watertown, MA, USA)	This new generation InMotion ARM^®^ is an evidence-based neurorehabilitation technology that provides patients with real-time Assistance-as-Needed™. The system is used for intensive motor therapy, consisting of two robots that work together to train to reach and grasp and release movements at the same time. The device quietly monitors the patient’s movements during therapy while it gently assists where needed to help them complete various motor therapy activities.
	ARMOTION (Reha Technology AG, Switzerland)	Armotion is a robotic solution in the treatment of severe and moderate neuromuscular dysfunction of the upper extremity. It maximizes the patient’s capability to undertake personal self-care and domestic tasks by rebuilding functional ability and the required skills for independence in daily life. It allows engaging and repeatable exercises with video feedback/videofeedback in a 2D workspace, data collection, and reporting and accurate patient performance assessment.
	MOTORE (Humanware, Pisa, Italy)	MOTORE “Mobile robot for upper limb neurOrtho Rehabilitation” is designed to provide rehabilitation exercises and to measure the person’s progress and performance. It is equipped with motors that actively assist movement to obtain the best rehabilitation: the system supports or opposes movement according to patients’ needs.
	ReoGo™ (Motorika, NJ, USA)	REOGO is a motorized and ergonomic robotic arm, which combines personalized, patient-specific exercises and engaging games. Motorika’s ReoGo™ enables two-or three-dimensional movements, allowing patients who have suffered a stroke or other neurological injuries to essentially retrain the brain through measured repetitive motion and video feedback.

**Table 2 jcm-10-05245-t002:** Summary of clinical/demographic characteristics (mean ± SD) of the groups at the baseline (T0).

	Gender/*n*	Age (Years)/*n*	Time Since Stroke/*n*	FMA-UE	MI	B&B	FAT	MAS			NRS	BI
								Shoulder	Elbow	Wrist		
Exoskeleton (*n* = 65)	M/37	<50: 12	early-subacute: 17	21 ± 5	41 ± 10	56 ± 13	1.3 ± 0.3	0.6 ± 0.2	1.1 ± 0.3	0.6 ± 0.2	1.6 ± 0.4	22 ± 5
		50–70: 32	late-subacute: 26	17 ± 3	38 ± 7	47 ± 09	2 ± 0.4	0.6 ± 0.1	0.9 ± 0.2	1 ± 0.2	3.3 ± 0.6	34 ± 7
	F/28	>70: 21	chronic: 22	19 ± 4	45 ± 9	55 ± 12	1.5 ± 0.3	0.7 ± 0.1	1.2 ± 0.2	0.8 ± 0.2	3.6 ± 0.8	52 ± 11
End-effector (*n* = 40)	M/22	<50: 6	early-subacute: 26	19 ± 4	48 ± 9	55 ± 11	1.5 ± 0.3	0.7 ± 0.1	1.1 ± 0.2	0.8 ± 0.2	3.7 ± 0.7	53 ± 10
		50–70: 19	late-subacute: 7	35 ± 13	58 ± 22	94 ± 36	1.9 ± 0.7	1.5 ± 0.6	1.5 ± 0.6	1.2 ± 0.4	1.1 ± 0.4	44 ± 17
	F/18	>70: 15	chronic: 7	29 ± 11	37 ± 14	52 ± 2	1 ± 0.4	1 ± 0.4	1.3 ± 0.5	1.5 ± 0.5	2.6 ± 1	51 ± 19

Legend: number, *n*; male, M; female, F; Fugl-Meyer Assessment for the upper extremity, FMA-UE; Motricity Index, MI; Box & Block Test, B&B; Numerical Rating Scale, NRS; Frenchay Arm Test, FAT; Barthel Index, BI; Modified Ashworth Scale, MAS; standard deviation, SD.

**Table 3 jcm-10-05245-t003:** Means and standard deviation of the outcome measures.

		T0	T1	T1–T0 *p*-Value	T2	T2–T0 *p*-Value
Exoskeleton	FMA-UE	19 ± 2	30 ± 4.5	<0.0001	39 ± 5	<0.0001
	MI	41 ± 5	61 ± 8	<0.0001	60 ± 7	<0.0001
	B&B	53 ± 7	65 ± 8	<0.0001	75 ± 9	<0.0001
	FAT	1.6 ± 0.2	2.9 ± 1.1	<0.0001	3.5 ± 2	<0.0001
	MAS shoulder	0.6 ± 0.1	2 ± 0.4	<0.0001	1.4 ± 0.5	<0.0001
	MAS elbow	0.7 ± 0.4	1 ± 0.1	<0.0001	1 ± 0.1	<0.0001
	MAS wrist	0.7 ± 0.3	0.8 ± 0.1	0.01	1 ± 0.1	<0.0001
	NRS	2.8 ± 0.4	2 ± 0.5	<0.0001	1 ± 0.1	<0.0001
	BI	36 ± 5	45 ± 6	<0.0001	48 ± 7	<0.0001
End-effector	FMA-UE	28 ± 5	34 ± 7	<0.0001	47 ± 7.5	<0.0001
	MI	48 ± 7.5	71 ± 11	<0.0001	75 ± 12	<0.0001
	B&B	67 ± 11	62 ± 4	0.0008	72 ± 4	0.0008
	FAT	1.5 ± 0.2	2.6 ± 1.5	<0.0001	3.6 ± 1.2	<0.0001
	MAS shoulder	0.9 ± 0.2	2 ± 0.6	<0.0001	1.5 ± 0.8	<0.0001
	MAS elbow	1.3 ± 0.2	1 ± 0.1	<0.0001	1 ± 0.1	<0.0001
	MAS wrist	1.2 ± 0.2	1 ± 0.1	<0.0001	1 ± 0.1	<0.0001
	NRS	2.5 ± 0.4	2 ± 0.4	<0.0001	1.5 ± 0.5	<0.0001
	BI	49 ± 8	59 ± 9	<0.0001	62 ± 10	<0.0001

Legend: Fugl-Meyer Assessment for the upper extremity, FMA-UE; Motricity Index, MI; Box & Block Test, B&B; Numerical Rating Scale, NRS;Frenchay Arm Test, FAT; Barthel Index, BI; Modified Ashworth Scale, MAS; baseline, T0; post-treatment, T1; three-month follow-up, T2.

**Table 4 jcm-10-05245-t004:** Robotic device-specific aftereffects.

		Time Effect (F, *p*)	T1–T0	T2–T0
Exoskeleton *n* = 65	Armeo Power *n* = 42	83, <0.0001	<0.0001	<0.0001
	Armeo Spring *n* = 23	76, <0.0001	0.001	0.001
End-effector *n* = 40	InMotion^®^ 2.0 *n* = 9	85, <0.0001	<0.0001	<0.0001
	ARMOTION *n* = 6	75, <0.0001	0.002	<0.0001
	MOTORE *n* = 17	26, <0.0001	0.003	<0.0001
	REOGO *n* = 8	15, <0.0001	0.015	<0.0001

## Data Availability

The data presented in this study are available on request from the corresponding author.

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
