# Peer review of "Robot-Assisted Training for Upper Limb in Stroke (ROBOTAS): An Observational, Multicenter Study to Identify Determinants of Efficacy"

_jcm, 2021, doi:10.3390/jcm10225245_

Round 1

Reviewer 1 Report

Robotic therapy is a promising technology for rehabilitation of patients with motor disorders caused by stroke.  Calabrò et al investigate the efficacy of the end-effector and exoskeleton types of robot devices in different phase of post-stroke in a prospective study, providing some specific information in the selection of both devices. This study was well-organized and conducted. However, some important inforamtion and detailed statistic data are not shown properly. Also some concerns need to be addressed.

  1. Whether the subjects received the conventional therapy during the study?
  2. Were the subjects outpatients or inpatients?
  3. Table2: Each row represents different stroke phase but not age. Age information should be in the second column before stroke phase column. Recent table is easy to confuse readers. It appears the early-sub-acute group contained 12 patients whose age <50.
  4. Line 177 author said “Baseline characteristics are summarized in Table 2”. Whether the score in table2 was got from T0 phase? Clear information needed.
  5. How to explain the inconsistence in table2 and 3? For an example, in table2 FMA-UE in exoskeleton group was 21, 17, 19 but FMA-UE T0 was 29 in table 3.
  6. Fig2: what dose each color mean? Was error bar SD or SE?
  7. Fig2: how to calculate the FMA-UE gain?
  8. Fig2: FMA-UE here was from all the patients or the patients whose FMA-UE higher than the MCID. Need clarification.
  9. The statistic information needs to be shown in table3 not just partially shown in the result section.
  10. Table 4: n, number for each group needs to be provided.
  11. Line192-195 figures needed for those results.
  12. Line200-206: no related information found in figure2. The authors had better make a table to show the detailed information (patients % higher than the MCID and FMA increase points from T0-T1 in different phase in two groups)

Author Response

We thank the reviewer for the appreciation of our manuscript and the useful suggestions to improve its quality. Please, find all details in the revised version of the manuscript (in track changes modality) and the point-to-point response to the comments.

  1. Whether the subjects received the conventional therapy during the study?

Yes, all the subjects received conventional training, as specified at the end of the rehabilitation paradigm.

  1. Were the subjects outpatients or inpatients?

They were inpatients, as better specified in the main text.

  1. Table2: Each row represents different stroke phase but not age. Age information should be in the second column before stroke phase column. Recent table is easy to confuse readers. It appears the early-sub-acute group contained 12 patients whose age <50.

According to reviewer’s suggestion, the age column was put before the stroke phase one.

  1. Line 177 author said “Baseline characteristics are summarized in Table 2”. Whether the score in table2 was got from T0 phase? Clear information needed.

We specified that the summary of clinical-demographic characteristics of the groups in table 2 referred to the baseline (T0) values.

  1. How to explain the inconsistence in table2 and 3? For an example, in table2 FMA-UE in exoskeleton group was 21, 17, 19 but FMA-UE T0 was 29 in table 3.

We thank the reviewer for having highlighted these inconsistencies and we apologize for these mere typing errors of averages and rounding from the three subgroups per stroke-phase to the overall average. Data were revised and corrected where appropriate.

  1. Fig2: what dose each color mean? Was error bar SD or SE?

Fig. 2 was redrawn since misleading. We now better pointed out the changes of FMA-UE at the different time points. The vertical error bars refer to SD (as now specified).

  1. Fig2: how to calculate the FMA-UE gain?

We better specified how FMA-UE gain was calculated (lines 201-208 of the manuscript_TC version).

  1. Fig2: FMA-UE here was from all the patients or the patients whose FMA-UE higher than the MCID. Need clarification.

The data in figure 2 refer to all the patients.

  1. The statistic information needs to be shown in table3 not just partially shown in the result section.

Thank you for the suggestion. We added the missing information.

  1. Table 4: n, number for each group needs to be provided.

We added the missing information, as requested. 

  1. Line192-195 figures needed for those results.

As suggested, we added a figure (now fig.2) illustrating the overall FMA-UE changes in the different subgroups.

  1. Line200-206: no related information found in figure2. The authors had better make a table to show the detailed information (patients % higher than the MCID and FMA increase points from T0-T1 in different phase in two groups)

The required information is now outlined in the new figure 2.

Reviewer 2 Report

Overall, very good efforts by the authors. This is a really good observational study which provides a ground work for more robust, standardized and structured future studies. I do not have any high level comments. Some minor suggestions as follows: 

  1. Please use complete word for FMA-UE in the abstract and in the article when uses for the first time, even for the very commonly used acronyms.
  2. Please provide citation for line 55-56
  3. Please use consistent form of citation in line 66.
  4. Line 84 – Please elaborate what are these specific generated devices.
  5. Line 115-117: Did authors use any criteria in allocation of exoskeleton vs end-effector device? Please specify in the article.
  6. Line 170 - Please provide more information for what consists of achievement of MCID for FMA-UE in your study, any particular number?

Thank you

Author Response

We thank the reviewer for the appreciation of our manuscript and the useful suggestions to improve its quality. Please, find all details in the revised version of the manuscript (in track changes modality) and the point-to-point response to the comments.

  1. Please use complete word for FMA-UE in the abstract and in the article when uses for the first time, even for the very commonly used acronyms.

Thank you for this note. We have amended as requested.

  1. Please provide citation for line 55-56. 

The relative references were provided [1-2].

  1. Please use consistent form of citation in line 66.

We have amended as notified.

  1. Line 84 – Please elaborate what are these specific generated devices.

UL movements differently from lower limbs ones are not easy to emulate and reproduce by robots. We have better specified it in the manuscript. 

  1. Line 115-117: Did authors use any criteria in allocation of exoskeleton vs end-effector device? Please specify in the article. 

No allocations were provided as this study is about the real use of robotics in clinical practice.

  1. Line 170 - Please provide more information for what consists of achievement of MCID for FMA-UE in your study, any particular number?

We specified that the primary outcome measure was the rate of achievement of the Minimal Clinically Important Difference (MCID) in FMA-UE at the T1-T0 change, which consisted in the change from baseline of the FMA-UE of at least 10 points in the subacute phase and of at least 5 points in the chronic one after the treatment, in both groups.

Thank you